# Climate-Smart Agriculture and Non-Agricultural Livelihood Transformation

**Jon Hellin [1],\* and Eleanor Fisher [2]**

[1] Sustainable Impact Platform at the International Rice Research Institute (IRRI), Metro Manila 1301, Philippines

[2] School of Agriculture, Policy and Development at the University of Reading, Reading RG6 6AH, United Kingdom; e.fisher@reading.ac.uk

\* Correspondence: j.hellin@irri.org

**Abstract:** Agricultural researchers have developed a number of agricultural technologies and practices, known collectively as climate-smart agriculture (CSA), as part of climate change adaptation and mitigation efforts. Development practitioners invest in scaling these to have a wider impact. We use the example of the Western Highlands in Guatemala to illustrate how a focus on the number of farmers adopting CSA can foster a tendency to homogenize farmers, instead of recognizing differentiation within farming populations. Poverty is endemic in the Western Highlands, and inequitable land distribution means that farmers have, on average, access to 0.06 ha per person. For many farmers, agriculture per se does not represent a pathway out of poverty, and they are increasingly reliant on non-agricultural income sources. Ineffective targeting of CSA, hence, ignores small-scale farming households' different capacities for livelihood transformation, which are linked to the opportunities and constraints afforded by different livelihood pathways, agricultural and non-agricultural. Climate-smart interventions will often require a broader and more radical agenda that includes supporting farm households' ability to build non-agricultural-based livelihoods. Climate risk management options that include livelihood transformation of both agricultural and non-agricultural livelihoods will require concerted cross-disciplinary research and development that encompasses a broader set of disciplines than has tended to be the case to date within the context of CSA.

**Keywords:** climate-smart agriculture; livelihood transformation; Guatemala; climate change

## 1. Introduction

Climate change will have a detrimental impact on agricultural productivity in many parts of the developing world [1]. Farmers have long adapted to climate variability, but the severity of the predicted changes may be beyond many farmers' current ability to adapt and improve their livelihoods [2,3]. There is an urgent need to work with farmers to develop climate change adaptation, mitigation and transformation strategies. Sustainable development goal (SDG) 13 is on *Climate Action* and, hence, there is much interest in the promotion of climate-smart agricultural practices (CSA). These are practices that contribute to an increase in global food security (and other development goals), an enhancement of farmers' ability to adapt to a changing climate and the mitigation of emissions of greenhouse gases [3,4]. CSA, hence, not only contributes to the realization of SDG 13, but is also intrinsically linked to several other SDGs, for example, SDG 1: *No Poverty* and SDG 2: *Zero Hunger*.

Transformative approaches have gained traction in contemporary policy debates on climate impacts, stimulated, amongst other factors, by the United Nations Sustainable Development Goals (SDGs) and the Intergovernmental Panel on Climate Change (IPCC). CSA can be transformative in terms of its aims to ensure food security via a reorientation of agricultural development in the

context of the realities of climate change. For example, recent research on the climate-smart village (CSV) approach [4] highlights the potential of scaling out so as to benefit larger number of farmers. However, there has been limited scaling of the CSV approach. One of the challenges is that the scaling of the CSV approach is premised largely on identifying a portfolio of CSA options and the financial or institutional mechanisms that enhance adoption by farmers, and targeting these at regions with similar agro-ecological conditions [4], with less attention being given to the local context [5,6]. The danger is the a priori belief that CSA is a pathway out of poverty. For many farmers, adaptation to climate change in ways that lead to an escape from poverty, and greater prosperity may not be via CSA [7].

Lipper et al [3] stress that CSA results in higher resilience and lower risks to food security. While this may be the case, there are farmers for whom agricultural-based livelihoods are so precarious that even "climate-proofing" their agricultural systems represents a higher risk to food security and prosperity than non-agricultural livelihood options. The challenge, therefore, is that at the same time that international calls for transformative approaches are made, current and future rural livelihood conditions are so adverse that, for some, this is a matter of changing to grasp any livelihood opportunity, including adverse coping strategies, without any ability to improve agricultural practices in ways that could be considered synonymous with transformative change in a positive sense. Hence, "climate-smart" may actually mean the need for actions that focus on supporting people in building non-agricultural-based livelihoods [3,4]. If this livelihood transformation is to be positive, it will require concomitant policy and development support to provide enabling conditions for non-agricultural livelihoods to be built. Moreover, this needs to be performed in ways that improve household income and security, i.e. are prosperity-enhancing, thus avoiding recourse to adverse coping strategies. The Western Highlands of Guatemala illustrates this challenging development scenario.

## 2. Climate-Smart Agriculture in the Western Highlands, Guatemala

Scientific evidence points to negative impacts on agriculture in Guatemala, and other parts of Central America, due to changing temperature and rainfall patterns, e.g., [8]. Inequalities in land distribution have forced many resource-poor farmers to farm steep hillsides, areas that are very susceptible to soil and land degradation. The response has often been the promotion of CSA. Development practitioners are rediscovering technologies and practices that were promoted in the region in the 1980s and 1990s under the guise of soil and water conservation [9]. These included live barriers, stone terraces, cover crops, green manures and agro-forestry. Farmers' uptake of these technologies and practices was disappointing 20–30 years ago [10], largely because, as is the case worldwide, a technology-led approach tends to ignore the needs for institutional enabling factors, which are very important when it comes to farmers' uptake of agricultural technologies [6,11,12].

The promotion of CSA in Guatemala is particularly challenging. The country suffers from extreme rural poverty and food insecurity [13]. Guatemala is ethnically very diverse, and indigenous groups (who make up almost 40% of the total population) live mainly in the Western Highlands. The underpinnings of present-day poverty are rooted in conflict, linked to Guatemala's 36-year civil war, which ended in the mid-1990s, and during which tens of thousands of indigenous people died [14]. This has left a legacy of inequality and continued social tension.

Small-scale farmers practice largely subsistence and some market-oriented agriculture. The most important cultivated food crop is maize, which is intercropped with beans, chilies and squash [15]. Recent research has shown that land availability in the Western Highlands is 0.06 ha per person [13]. This contributes to considerable food insecurity: farm households produce enough maize (the main staple crop) for fewer than seven months of consumption per year, and for household consumption have to purchase maize to make up the deficit. As a consequence, the majority of farmers seek off-farm employment on a temporary basis, while a minority have managed to branch into the production of higher-value vegetable crops for the export market [16].

Donors have invested much in rural development projects [17,18]. One such rural development project in the Western Highlands was implemented from 2013–2018. The Buena Milpa project was

supported by the United States Agency for International Development (USAID), through its Global Hunger and Food Security Initiative "Feed the Future". Its main objectives were to reduce poverty, food insecurity and malnutrition, while increasing the sustainability and resilience of maize-based farming systems (The ideas reported here stem from Hellin's involvement as a socio-economist in this project in the Western Highlands of Guatemala). More details are provided in [13,19]. A strong emphasis of the project was the promotion of CSA, and the project worked through a number of non-governmental organizations. During the course of the work, it became clear that more attention needed to be focused on farmers' different capacities to engage in climate risk management. The danger was that poor targeting of CSA would lead to weak farmer uptake, by implication excluding many poor farmers and/or including those farmers for whom farming (and improvements in farm productivity via the use of CSA) would do little to enable them to escape poverty.

That project brought to the fore the need to recognize more explicitly the heterogeneity of farm households and the need to broaden the portfolio of livelihood options available to them. A further challenge was to accommodate the understandable desires of the donor to see an impact on the ground. There was pressure to scale CSA, in terms of enhanced farmer uptake of technologies and practices. Implicitly, this served to dismiss emerging evidence that the role of non-agriculture-based livelihoods needed to be taken into account in decision-making regarding appropriate interventions; the promotion of livelihood improvement through CSA was inappropriate for some categories of farmer. There was a danger that the focus on numbers would distract from whether farmer uptake of CSA, while contributing to an improvement in food security, would still leave farmers trapped in poverty, not to mention the potential for other unanticipated impacts, such as when wealthier farmers are able to capture the benefits of CSA, with the consequence that their wealth grows at the expense of poorer farmers, leading, ultimately, to greater social inequality.

The project in Guatemala clearly demonstrates the importance of priority-setting and factoring in the varied possibilities and local conditions that farmers face when it comes to targeting project interventions. Thornton et al [5] provide a useful framework for CSA priority-setting that is based on six elements, and is designed to help guide best-bet CSA intervention. There is widespread recognition of the trade-offs when implementing CSA among the three pillars of food security, adaptation and mitigation [5,6]. The example of the Western Highlands illustrates "higher-level" trade-offs between some of the SDGs. These include trade-offs between SDG 13: *Climate Action* and SDG 5: *Gender Equality* together with SDG 10: *Reduced Inequality*.

In short, the Guatemalan project illustrates that a focus on the number of farmers adopting CSA can divert attention from the far more important issue, which is to support farmers' adaptation to climate change, either through making their agriculture-based systems more climate-resilient and/or by expanding their envelope of prosperity-enhancing non-agricultural livelihoods. The latter has been less prevalent in CSA interventions, and this has been at the expense of potentially ensnaring poorer categories of small-scale farmers in an agricultural-based poverty trap.

## 3. Climate-Smart Agriculture and Poverty Reduction

Farm households can be distinguished based on their asset endowment, e.g., their amount of land, access to key agricultural inputs etc., coupled with characteristics that determine the livelihood strategies available to them. These livelihood strategies, in turn, influence the livelihood incomes that hopefully enable a household to maintain and strengthen its livelihood security. The livelihood pathways available to a farm household are determined by the household's characteristics (e.g., dependency ratio, availability of labour, etc.), along with the interaction between the available assets (financial, natural, social, human) and the enabling or disabling economic, institutional and policy environment. An understanding of these livelihood pathways informs decisions as to where to target CSA and where to develop enabling approaches that facilitate livelihood changes.

There is no doubt that CSA and agricultural interventions can contribute to poverty reduction and enhanced prosperity. Numerous examples abound, e.g., [20–22]. However, the agricultural future is bleak for some farmers struggling with few resources and the additional challenge of climate change. Harris and Orr [23] argue that for rain-fed agriculture, crop production could be a

pathway from poverty where smallholders are able to increase farm size or where markets stimulate crop diversification, commercialization and increased farm profitability. The potential to improve productivity is also, of course, important. Nevertheless, as Cavanagh et al [24] comment, "*the poor and less poor are […] more capable of diversifying into off-farm and non-farm activities compared to the very poor, whose small land holdings and poor access to capital constrain their ability to diversify away from on-farm income and seasonal off-farm wage labour*". We certainly found this to be the case in the Western Highlands of Guatemala [13].

Agriculture is not a pathway out of poverty for all farm households. Hence, for certain categories of household, poverty reduction will come from farmers moving out of agriculture and into the non-farm economy. For poor households, non-agricultural livelihood transformation can, of course, represent nothing more than a negative coping strategy. The challenge is to ensure that non-agricultural livelihood options are positive, i.e., prosperity-enhancing. It is a challenge in all parts of the world due to profound rural changes. In many parts of the world, agricultural production will have to increase hugely, along with labour productivity; the latter will lead to fewer people engaging in agriculture [25]. This has already led, in parts of Asia, to what Li [26] refers to as a rural population that is "surplus" to the needs of capital, as many of those dispossessed from their land are also unable to find meaningful employment off-farm. It is also increasingly common in Latin America.

The idea of a "surplus" population mirrors the earlier thesis of "functional dualism", proposed by de Janvry et al [27] and expanded on by Blaikie [28]. The authors suggest that farmers rely upon returns from market activities to complement their agricultural returns from farming plots of land that are too small to allow for self-sufficiency. Farmers are often obliged to work as part-time wage laborers due to their resulting food insecurity, thus needing to make up shortfalls of staples and cash requirements for household goods, as well as to pay for inputs for the production process itself on their farms. They are increasingly dependent on non-farm sources of income but are unable to find sufficient employment opportunities or capital to migrate (and abandon the agricultural sector) or to depend fully on wage earnings for their subsistence. Returns from subsistence-oriented agricultural activities provide a necessary complement to the low wages that farmers receive in the labour market. In addition, where opportunities for wage labour are primarily in the agricultural sector, poor returns from own-farm agricultural production are reinforced, given that peak demand for agricultural labour may coincide with labour demands on farmers' own land.

The situation in the Western Highlands of Guatemala, as described in the section above, is in keeping with the functional dualism thesis, and has major implications for identifying and targeting appropriate pathways leading to rural poverty reduction. As suggested, for many farmers in the Western Highlands, CSA may not be an attractive option because of labour and land shortages. In the case of many farmers in the Western Highlands of Guatemala, temporary migration in search of non-farm employment has been a traditional coping strategy, with farmers investing the earned off-farm income in their villages and/or diversifying into non-farm agricultural activities, such as setting up a local shop. For many farmers, labour, essential for investment in soil improvement or maintenance of conservation structures, is not available, because they are working off-farm (and households may also have high dependency ratios) [23]. Similarly, another refrain, when CSA practices such as conservation agriculture are promoted, is that farmers should not burn their fields to clear the vegetation prior to planting because of the adverse impact on soil quality, especially biological health. For farmers who have been working off-farm, and for whom labour is scarce, this can be an unattractive recommendation.

CSA is also very problematic when it comes to small-scale farmers with very small landholdings, as is the case in the Western Highlands. Firstly, in the case of cross-slope soil conservation technologies, such as live-vegetation barriers and stone walls, land is taken out of production. In the case of live barriers, however, this can be partly compensated for by using species that make a contribution to the farm household, e.g., edible products for humans and/or animals. Secondly, even if CSA were to lead to significant improvements in agricultural productivity, the increase (while a contribution to food security) would be unlikely to help the farmer escape poverty.

"*For most smallholders, however, small farm size and limited access to markets mean that returns from improved technology are too small for crop production alone to lift them above the poverty line*" [23].

This raises the question of how best to support categories of farmers who are being targeted but whose small holdings, household structure and asset endowment may be inappropriate for the measures advocated under the guise of CSA. It may be the case that the CSV approach would be more successful, but in the absence of effective scaling of this approach and comprehensive impact studies, this remains an under-researched area.

## 4. Non-Agricultural Livelihood Transformation

In rural contexts where small-holder farmers are based, development involves decreasing livelihood vulnerability and increasing incomes, typically through changes in livelihood activities [29]. Ideally, CSA should enable farmers to pursue livelihood pathways that lead to greater prosperity, while also building resilience. Recent thinking has advocated addressing the need to support changes that can be transformative, in the face of climate-related impacts that imply dramatic changes to environmental conditions. There is much research on developing a framework for assessing and comparing different types of interventions that address the key elements of CSA [5,10]. This research is necessary and important; however, in the context of agricultural transformation, the focus needs to broaden to systematically factor in livelihood trajectories outside of agriculture. CSA, to enhance food security and meet the SDGs, will require a longer-term perspective and bolder action that comprehensively targets farmer livelihoods [5].

The reality is that positive, sustainable livelihood pathways within the agricultural sector may not be an option for all types of farmers, i.e., not all households face the agro-ecological and socio-economic conditions necessary to move from one asset threshold and livelihood pathway to another, enabling them to escape poverty, while still remaining in agriculture. Guatemala epitomizes this reality. A report produced for the United States Agency for International Development (USAID) noted that "*given the agricultural foundation and 'capital' that many Western Highland communities continue to hold, [there is a need to] re-assess the productive options available in agriculture or agriculture-based livelihoods; and engage youth (many of whom have written agriculture off as an option) in development of potential integrated economic/environmental/social development initiatives*" [18].

Incorporating non-agricultural livelihood transformation within CSA requires innovative and open thinking on the way forward for CSA. This has been acknowledged by proponents of CSA, e.g., [3,4], but it poses disciplinary challenges and has not led to the type of holistic and transformative changes are that needed. What is required is a broader and more comprehensive understanding of the realities faced by farmers and the changes needed to foster large-scale transformation in their livelihood trajectories [30]. This means that CSA thinking has to involve those from a plethora of disciplines from the natural and social sciences [31]. In the context of CSA, we have a practical example of how transformation also becomes a political issue [32]. The debate around adaptation to environmental change often avoids questioning the socio-economic and political reasons why farmers' livelihoods are so vulnerable [33].

In the context of Guatemala, serious discussion around climate change adaptation, mitigation and transformation will have to contend with politically divisive issues. In a small way, within a project, "politics" can mean challenging the premises that drive inappropriate scaling. Within the bigger picture, it also means taking into account the political economies within which CSA is advocated for small-scale farmers. In the Guatemalan context, this political economy relates to several decades of conflict and on-going socio-economic inequality that structurally disadvantage small-scale farmers in the Western Highlands. While there are, of course, specificities to the political economy of Guatemala, which have shaped its uptake of CSA, in any given context there will be political economy issues underpinning the implementation of CSA that cannot and should not be ignored. The climate change discourse has tended to focus on the adaptation and mitigation of greenhouse gas emissions, rather than "*problems of unevenly distributed power relations, networks of control and influence, and rampant injustices of the 'system*" [34].

Clearly, a disregard for issues of power and inequality is not tenable if CSA is to provide a viable mechanism for livelihood transformation and a contribution to the SDGs. Indeed, there is growing evidence of CSA proponents adopting a more "radical" agenda, factoring in more readily political and institutional issues and ensuring that CSA debate and implementation does not remain largely a discourse among "elite development and research agencies" [35]. Such recognition of the political realities of small-holder agricultural development are important if CSA is to have continued longevity and relevance within international agendas on climate change action and the SDGs.

## 5. Conclusions

Climate adaptation requires transformative change. The CSA approach needs to move even more squarely beyond a focus on resilience of food systems to encompass systematic thinking and action with respect to the resilience of farm households. This poses a real challenge, because CSA has tended to overlook targeting issues related to socio-economic differentiation within small-scale farming populations, although recent CSA initiatives have more readily included analyses of the institutional dimensions of CSA. Greater acknowledgement of institutional issues, and indeed the politics, of CSA interventions within rural planning are to be welcomed. CSA has nevertheless, in practice, tended to exclude systematic consideration of support for non-agricultural livelihood transformation that is positive for farm households in marginal contexts, such as the Western Highlands of Guatemala.

In some cases, CSA can lead to the triple win of increased productivity, adaptation and mitigation, but this is not the case for all types of farmers. We argue that more systematic attention be directed at climate risk management that moves beyond the more conventional adaptation and mitigation discourse, towards an approach that includes livelihood transformation from a broader perspective, i.e., one that does not just focus on rural–agricultural transformation, but also identifies (and embraces) where agriculture per se is not a pathway out of poverty and where support for positive non-agricultural livelihood trajectories are needed for small-scale farmers. This requires more disciplines working together, and, perhaps, meeting the challenge of addressing entrenched power balances, both within communities of scientists and in the small-scale farming populations that are the subject of CSA interventions.

**Author Contributions:** Conceptualisation, J.H. and E.F.; writing—original draft preparation, J.H. and E.F.; writing—review and editing, J.H. and E.F.

**Funding:** The research in Guatemala reported here was funded by the United States Agency for International Development (USAID) through its Global Hunger and Food Security Initiative, 'Feed the Future'. This work was also supported by the CGIAR Research Program (CRP) on Rice Agri-food Systems (RICE, 2017-2022) and the CRP on Climate Change, Agriculture and Food Security (CCAFS), which is carried out with support from CGIAR Fund Donors and through bilateral funding agreements (for details please visit https://ccafs.cgiar.org/donors#.WxqT_4onaUk). The views expressed in this document cannot be taken to reflect the official opinions of the aforementioned organizations.

**Acknowledgements:** The authors would also like to thank two anonymous reviewers who provided invaluable comments on earlier versions of this paper.

**Conflicts of Interest:** The authors declare no conflict of interest. The funders had no role in the design of the study; in the collection, analyses, or interpretation of data; in the writing of the manuscript, or in the decision to publish.

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
