# Peer review of "Climate-Smart Agriculture and Non-Agricultural Livelihood Transformation"

_climate, doi:10.3390/cli7040048_

Round 1
Reviewer 1 Report
Summary
Thank you for the opportunity to review this paper. The main argument made by the authors is an important one. Specifically, Hellin and Fisher advocate for proponents for Climate Smart Agriculture (CSA) broaden their scope to include non-agricultural livelihood transformation. The authors did a good job connecting this premise with multiple Sustainable Development Goals. They argue that CSA is an approach that can be applied to reaching the SDGs, in some cases. The premise of the paper is that the goals of CSA should include the well-being of land-based communities in rural and resource-poor areas. However, I believe the authors must do more to convince readers that this is true. As I currently read the paper, the authors are evaluating CSA based on standards that it was not designed to meet. This is, unfortunately, a fundamental flaw that undermines the manuscript as a whole. One option for addressing this flaw is to find a different development framework that justifies the livelihoods prerogative, and use a suite of initiatives (including by extending beyond CSA) to evaluate the current conditions in the Western Highlands.
Major issues
1. I find that the authors have overstated certain conditions. For example, they write: “While it is true that famers have long adapted to climate variability, predicted changes are so severe that they are likely to be beyond many farmers’ current ability to adapt and improve their livelihoods.” (lines 34-36) Only the first part of this statement is faithful to the citation source (Adger et al), and there is little evidence to support the second sentence that I am aware of. There are several problematic overstatements like this throughout, which should be better cited to improve the credibility of the manuscript.
2. The case study approach could be better developed and organized. As someone who has never had the opportunity to go to the Guatemalan Western Highlands, I found myself wanting a better study site description. How many people, what types of agriculture, what is the poverty rate, how prevalent is food insecurity, and what time of year is hunger most problematic? What research has been done on agricultural livelihoods in this area? How long have CSA advocates been working in this region? What were the (apparently unsatisfactory) attempts to promote conservation agriculture before CSA came along, and who supported them? Who is supporting CSA practices now, and are they measuring success in any way. As a whole, the case study is not as well connected to the paper’s thesis as it should be.
3. There is a large body of literature on livelihood diversification that should be included in this review.
4. I would suggest that the authors include a standard introduction in this paper.
Minor issues
1. Inconsistent use of capitalization in “Climate Smart Agriculture”.
2. Inconsistent tense usage throughout.
3. Several long run on sentences, or sentences whose meaning are unclear.
4. The authors should better differentiate when they are writing about agriculture at a global scale versus a regional or local scale.
5. Section headings do not always reflect what is in the sections
6. The authors may consider summarizing their discussion of why farmers do not like certain CSA practices in a descriptive table.
7. The authors need to cite or better justify the statement “The future would seem bleak especially for smallholder farmers…” (lines 122-123)
8. Several quotations feel plopped into the narrative without sufficient context setting.
Author Response
The reviewer made some very pertinent observations.
With respect to the observation that "the authors are evaluating CSA based on standards that it was not designed to meet" we have (in line also with the second reviewer's comments) revised the paper so that it acknowledges the relevance of CSA and the growing body of research that is addressing the issue of targeting. The argument of our paper is not to question CSA per se but to encourage proponents of CSA to broaden the discourse around the institutional framing of CSA and to acknowledge that for some farmers CSA is not a pathway out of poverty. In these cases 'climate smartness' includes facilitating these farmers' transition to alternative non-agricultural livelihoods.
MAJOR ISSUES
1. We have revised the manuscript to address the observation that in the manuscript we have overstated certain conditions e.g. lines 35-37.
2. We have expanded considerably on the Guatemala case study (Section 2) in order to provide the reader more information on the context of climate smart agricultural & development efforts in the Western Highlands.
3. Revisions in section 3 and 4 draw on the large body of research on livelihood diversification.
4. We have also re-written and shortened the Introduction as per the reviewer's comment.
MINOR ISSUES
1. We have standardized use of “Climate Smart Agriculture”.
2. We have made the use of tenses more consitsent.
3. We have revised the grammar so as to avoid long sentences.
4. we have sought to differentiate when writing about agriculture at different scales.
5. We have ensured that the section headings reflect content of the section.
6. The changed emphasis on the paper (see above) means that the paper is not about farmers' acceptance of CSA per se rather the need to broaden the whole climate smart approach to factor in more readily where non-agricultural transformation is needed. We have, hence, decided not to summarize why farmers do not like certain CSA practices in a descriptive table.
7. As per # 1 in the major issues (see above) we have revised the text so as to avoid these overstatements.
8. We have sought to provide more context when using the quotations.
Reviewer 2 Report
In the manuscript entitled “Climate smart agriculture and non-agricultural 2 livelihood transformation”, the authors argue that for CSA options, particularly implemented through projects, to have a wider impact, a transformative and interdisciplinary approach is required with more emphasis on non-agricultural livelihoods options. The authors contend that socioeconomic realities and conditions of smallholder famers are highly heterogeneous, even in the same environment. Thus, focusing on CSA technological options alone might not a viable pathway out of poverty for all farmers; for some farmers, non-agricultural livelihood transformation might be a more effective pathway out of poverty and to build resilience. They used their own research experience in Guatemala to illustrate their arguments.
The call for more emphasis on institutional enabling factors to facilitate out/up scaling of CSA is not new and has been highlighted in a recent systematic review paper which analyzed how institutional dimensions have been reflected in global CSA research. Research on targeting CSA options to the right context has been evolving recently and a framework for priority-setting in climate smart agriculture research across scales has been proposed. The climate-smart village (CSV) approach as technological and institutional options for dealing with climatic variability and climate change in agriculture has been tested in several regions around the world, including in Guatemala in response to the call for more attention to institutional and policy contexts of implementing CSA.
It would be good if the authors acknowledge the past and current efforts and initiatives which attempt to make CSA more targeted to socioeconomic and institutional contexts where farmers operate. I provide in the PDF file some useful documents to that regards. I have also provided some minors edits in the text. Kindly download the PDF file to see more
Line 8: Abstract
the background part of the abstract is disproportionate in comparison to results/analysis/finding part of the abstract. The result/analysis/finding is presented in just one sentence. Since the manuscript is not a review paper, it will be good if authors reduce the background information in the abstract and give more insights from the case study (from the Western Highlands in Guatemala ) to illustrate why it is necessary to or how to move from a narrow CSA thinking/implementation to a holistic livelihood transformation (including both agricultural and non-agricultural) approach for climate risk management
Line 52-52:
Not necessarily non-agricultural based livelihoods!
In essence CSA aims to achieve sustainable agricultural development for food security and is an approach for transforming and reorienting agricultural development under the realities of climate change. The authors might wish to read the paper below
§ Aggarwal, P. K., Jarvis, A., Campbell, B. M., Zougmoré, R. B., Khatri-Chhetri, A., Vermeulen, S. J., . . . Yen, B. T. (2018). The climate-smart village approach: framework of an integrative strategy for scaling up adaptation options in agriculture. Ecology and Society, 23(1). doi: 10.5751/ES-09844-230114
§ Thornton, P. K., Whitbread, A., Baedeker, T., Cairns, J., Claessens, L., Baethgen, W., . . . Keating, B. (2018). A framework for priority-setting in climate smart agriculture research. Agricultural Systems, 167, 161-175. doi: 10.1016/j.agsy.2018.09.009
§ Lipper, L., Thornton, P., Campbell, B. M., Baedeker, T., Braimoh, A., Bwalya, M., . . . Torquebiau, E. F. (2014). Climate-smart agriculture for food security. Nature Climate Change, 4(12), 1068-1072. doi: 10.1038/nclimate2437
Line 64-65:
This is valid for CSA implementation. A recent systematic review of institutional dimensions in global CSA research has shown that institutional enabling factors for CSA uptake have received less attention.
§ Totin, E., Segnon, A. C., Schut, M., Affognon, H., Zougmoré, R., Rosenstock, T., & Thornton, P. (2018). Institutional Perspectives of Climate-Smart Agriculture: A Systematic Literature Review. Sustainability, 10(6), 1990. doi: 10.3390/su10061990
Line 69:
Lessons learnt from CSA initiatives in West Africa indicate that institutional and policy setting at different scales (community, national to regional) are crucial to upscaling CSA technologies. See more in the paper below:
§ Partey, S. T., Zougmoré, R. B., Ouédraogo, M., & Campbell, B. M. (2018). Developing climate-smart agriculture to face climate variability in West Africa: Challenges and lessons learnt. Journal of Cleaner Production, 187, 285-295. doi: https://doi.org/10.1016/j.jclepro.2018.03.199
Line 70-71:
This has been emphasized in a recent systematic review paper that analyzed how institutional dimensions have been reflected in global CSA research. See more below
§ Totin, E., Segnon, A. C., Schut, M., Affognon, H., Zougmoré, R., Rosenstock, T., & Thornton, P. (2018). Institutional Perspectives of Climate-Smart Agriculture: A Systematic Literature Review. Sustainability, 10(6), 1990. doi: 10.3390/su10061990
Line 86-97:
The entry points for CSA intervention/implementation vary and range from the development of technologies and practices to elaboration of climate change models and scenarios, information technologies, insurance schemes, and processes to strengthen the institutional and political enabling environment. This diversity of possibilities and context-specificity of farmers underline the importance of priority-setting. See more in the paper below
§ Thornton, P. K., Whitbread, A., Baedeker, T., Cairns, J., Claessens, L., Baethgen, W., . . . Keating, B. (2018). A framework for priority-setting in climate smart agriculture research. Agricultural Systems, 167, 161-175. doi: 10.1016/j.agsy.2018.09.009
Second: the contexts of framing CSA is important. The underlying issues constructing the discourse around CSA differ. See more in the papers below
§ Chandra, A., McNamara, K. E., & Dargusch, P. (2018). Climate-smart agriculture: perspectives and framings. Climate Policy, 18(4), 526-541. doi: 10.1080/14693062.2017.1316968
§ Totin, E., Segnon, A. C., Schut, M., Affognon, H., Zougmoré, R., Rosenstock, T., & Thornton, P. (2018). Institutional Perspectives of Climate-Smart Agriculture: A Systematic Literature Review. Sustainability, 10(6), 1990. doi: 10.3390/su10061990
Line 97-101:
and yes there are trades-offs even among the 3 CSA pillars when implementing CSA which need to be addressed. See more
§ Totin, E., Segnon, A. C., Schut, M., Affognon, H., Zougmoré, R., Rosenstock, T., & Thornton, P. (2018). Institutional Perspectives of Climate-Smart Agriculture: A Systematic Literature Review. Sustainability, 10(6), 1990. doi: 10.3390/su10061990
§ Thornton, P. K., Whitbread, A., Baedeker, T., Cairns, J., Claessens, L., Baethgen, W., . . . Keating, B. (2018). A framework for priority-setting in climate smart agriculture research. Agricultural Systems, 167, 161-175. doi: 10.1016/j.agsy.2018.09.009
Line 112: Section 2. Climate-smart agriculture and poverty reduction
while barriers to upscaling CSA have been highlighted by authors in reference to their case study context, it is also import to acknowledge the success and cases where CSA approach has been effective in resilience building and poverty reduction. The papers below can serve as a starting point
§ Rosenstock, T. S., Lamanna, C., Namoi, N., Arslan, A., & Richards, M. (2019). What Is the Evidence Base for Climate-Smart Agriculture in East and Southern Africa? A Systematic Map. In T. S. Rosenstock, A. Nowak, & E. Girvetz (Eds.), The Climate-Smart Agriculture Papers: Investigating the Business of a Productive, Resilient and Low Emission Future (pp. 141-151). Cham: Springer International Publishing.
§ Dinesh, D., Frid-Nielsen, S., Norman, J., Mutamba, M., Loboguerrero, A. M., & Campbell, B. M. (2015). Is Climate-Smart Agriculture effective? A review of selected cases. CCAFS Working Paper no. 129
Line 131-132:
this is highly debatable though! Need to be rephrased to acknowledge that agricultural production can contribute to poverty alleviation too! Ideally agricultural and non-agricultural based approach is the best way.
There are several and strong evidence that support the opposite, ranging from theoretical to empirical findings. See a few (far from exhaustive) papers below:
§ Kassie, M., Shiferaw, B., & Muricho, G. (2011). Agricultural Technology, Crop Income, and Poverty Alleviation in Uganda. World Development, 39(10), 1784-1795. doi: https://doi.org/10.1016/j.worlddev.2011.04.023
§ Li, E., Deng, Q., & Zhou, Y. (2019). Livelihood resilience and the generative mechanism of rural households out of poverty: An empirical analysis from Lankao County, Henan Province, China. Journal of Rural Studies. doi: https://doi.org/10.1016/j.jrurstud.2019.01.005
§ Irz, X., Lin, L., Thirtle, C., & Wiggins, S. (2001). Agricultural Productivity Growth and Poverty Alleviation. Development Policy Review, 19(4), 449-466. doi: 10.1111/1467-7679.00144
§ Verkaart, S., Munyua, B. G., Mausch, K., & Michler, J. D. (2017). Welfare impacts of improved chickpea adoption: A pathway for rural development in Ethiopia? Food Policy, 66, 50-61. doi: https://doi.org/10.1016/j.foodpol.2016.11.007
Line 162:
Climate-smart village approach might be more relevant to those farmers than focusing narrowly on CSA technologies. See below a paper which documents the experiences and lessons, including barriers in implementing CSV approach in various locations and contexts in the world, including in Guatemala
§ Aggarwal, P. K., Jarvis, A., Campbell, B. M., Zougmoré, R. B., Khatri-Chhetri, A., Vermeulen, S. J., . . . Yen, B. T. (2018). The climate-smart village approach: framework of an integrative strategy for scaling up adaptation options in agriculture. Ecology and Society, 23(1). doi: 10.5751/ES-09844-230114
Line 173:
"Not burning fields" is not specific to CSA but it is about sustainability of agro-ecosystems and this mantra has been around well before the emergence of the concept of CSA, which is a fairly new concept.
Line 210-211:
this has been acknowledged in the Climate-Smart village approach. See
§ Aggarwal, P. K., Jarvis, A., Campbell, B. M., Zougmoré, R. B., Khatri-Chhetri, A., Vermeulen, S. J., . . . Yen, B. T. (2018). The climate-smart village approach: framework of an integrative strategy for scaling up adaptation options in agriculture. Ecology and Society, 23(1). doi: 10.5751/ES-09844-230114
Line 221-223:
See more here:
§ Chandra, A., McNamara, K. E., & Dargusch, P. (2018). Climate-smart agriculture: perspectives and framings. Climate Policy, 18(4), 526-541. doi: 10.1080/14693062.2017.1316968

Author Response
We would particularly like to thank reviewer 2 for his/her very helpful comments. As detailed below, we have taken on board all the comments and believe that the paper has improved substantially.
We have thoroughly revised the paper based on the suggestion that we make much more use of the literature on scaling of CSA and the scholars who have written on the importance of institutional enabling factors. In the attached document, you can see where we have referred to the work of almost all the papers that the reviewer referred to in his/her comments and also on the annotated PDF. Specifically we cite Dinesh et al 2015; Totin et al 2018; Thornton et al 2018; Partey et al. 2018; Chandra et al 2018; ; Lipper et al 2014 and Aggarwal et al 2018.
We have also referred to Kassie et al 2011 and Verkaart et al 2011 to correct the erroneous impression we gave in the original submission that farmers could not escape from poverty via agricultural pathways.
PLEASE SEE THE ATTACHED REVISED DOCUMENT IN TRACK CHANGES.
With reference to the second reviewer's specific comments:
With have revised the abstract to include more details on Guatemala.
Original line 52. Please see next text in lines 49-63 that address the reviewer's original comment about not focusing just on non- agricultural livelihoods.
Original lines 64-65. Please see revised text in lines 86-88.
Original lines 69 and 70-71. Please see reference (lines 96-97) to new literature as per reviewer's comments
Original lines 86-97. Revised text e.g. lines 161-167 refer to the CSA framework developed by Thornton et al 2018
The revised text (lines 161-168 also refer to the tradeoffs in CSA. the thrust of our argument is that there are also 'higher order' tradeoffs at play between the SDGs (please see revised text).
In the revised text e.g. lines 190-191, we acknowledge that CSA can be a pathway out of poverty for many farmers but not all. We cite the texts recommended by the reviewer.
In the revised text we have rephrased sections to reflect the argument that for some farmers agriculture is not a pathway our of poverty but for others it can be. As per above we cite work by Kassie et al etc. as recommended by the reviewer.
We refer to CSV e.g. lines 50-55 and also 240-241.
We have clarified the issue of burning of fields (original line 173) and now lines 252-253 in revised text.
Please see lines 318-325 that acknowledges the strides that have bene made to target CSA better.
We have reworked the conclusions.
Round 2
Reviewer 1 Report
Thank you for the opportunity to review this revision of the manuscript. I appreciate the authors' response to my suggestions. Specifically, they were successful in tweaking their framing, and better locating CSA as one strategy among several that can support climate adaptation within communities. I agree with their premise that livelihood strategies and food security issues need to be better incorporated into climate adaptation programming that targets rural agricultural communities.
I appreciate the addition of the Climate Smart Villages approach, and makes the point that CSA is just one approach among many. An additional reference like this would make the point more clearly, but in my opinion this is not necessary for the paper to be accepted.
The expanded description of the Guatemalan Western Highlands is very effective and strengthens the manuscript. I especially like how the authors conclude this section talking about the “higher level tradeoffs” of SDGs.
Lines 208-237 feel out of place. I suggest the authors think about how to better connect them with the first part of this section – probably just one or two more connecting sentences. Or the authors could consider moving them up in the section and ending on the broader perspective of why CSA techniques may not work for Western Highland farmers for reasons A, B, and C.
In my opinion, all 3 direct quotes are not necessary between lines 250-269. Consider rephrasing and citing unless there is simply no other way to say what you want to get across, or unless the source of the citation is pertinent to the point being made.
Lastly, it’s not a big deal at all (more of a stylistic suggestion): I hope the authors will consider slimming down on their use of per se. In some cases, it reads as if you are hedging when that isn’t necessary.
Overall, the manuscript is much improved.
Author Response
Again, we would like to thank the reviewer for his/her very useful comments. We have revised the paper as follows:
1. The reviewer wrote that "lines 208-237 feel out of place. I suggest the authors think about how to better connect them with the first part of this section – probably just one or two more connecting sentences. Or the authors could consider moving them up in the section and ending on the broader perspective of why CSA techniques may not work for Western Highland farmers for reasons A, B, and C.". IN RESPONSE, we re-wrote and re-ordered these sections as well as deleting the text about the author's experience in Honduras in the 1990s. Please see lines 180-204 in the attached revised manuscript
2. We have revised the text that originally appeared between lines 250-269. While we have kept the first direct quote, we have deleted the other two.
3. We agree with the reviewer that there was an over use of 'per se' and in the revised version we have deleted five of the six times that it appeared. We deleted it from lines 56, 126, 135, 178 and 250 of the second version of the manuscript.
Reviewer 2 Report
I have checked the revised manuscript and I am happy with most of the modifications made by the authors.
There are, however, some minor issues which need to be adequately addressed before consideration of the manuscript for publication.
Abstract:
The abstract still needs to be thoroughly revised. While the abstract is lengthy, it doesn’t sufficiently reflect the main arguments developed in the manuscript. The background part of the abstract is disproportionate in comparison to results/analysis/finding part of the abstract. The result/analysis/finding is presented in just one sentence.
I recommend that the authors reduce the background information in the abstract and give more insights from the case study to illustrate why it is necessary to or how to move from a narrow CSA thinking/implementation to a holistic livelihood transformation (including both agricultural and non-agricultural) approach for climate risk management. I highlight potential texts/parts that can be removed with strikethrough. Please consult the attached file.
Section 2. Climate smart agriculture in the Western Highlands, Guatemala
Please edit out the redundancy in this section in order to shorten it.
Acknowledgements: in the initial version of the manuscript, it was stated that the research depicted in this manuscript has been supported the CCAFS program. But in the revised version, the acknowledgement of CCAFS has just been edited out. Please could you explain why this discrepancy between the initial and the revised version?

Author Response
Once again, we would like to thank the reviewer for his or her very useful comments. We have taken these on board (see below) and once again the manuscript has improved.
1. An oversight on our part meant that we did not revise the abstract as thoroughly as we had intended. We have now down so and thank the reviewer for making this task so much easier by suggestions where these changes best be made.
2. The reviewer suggested a shortening of Section 2. Climate smart agriculture in the Western Highlands, Guatemala. However, another reviewer indicated that he or she appreciated the extra details on Guatemala. We have sought to steer a middle course by revising the section and reducing it from 1,455 words to 1,212. The reduction came from removing the redundant text.
3. The reviewer understandably questioned the change in the acknowledgements section with reference to the support provided by CCAFS. We grappled with the issue of the support of CCAFS. While revising the manuscript based on the reviewers' first comments, we realized that few of the ideas that we raise in the manuscript can be directly attributed to research supported by CCAFS (e.g. the project in Guatemala that we refer to was supported by USAID and was separate to CCAFS). However, the ideas that we put forward have evolved over the years in part through rich discussions with colleagues supported by CCAFS. We hope that the reviewer will be happy with the acknowledgements sections that now recognizes USAID and also the RICE (that supports the first author) and CCAFS CRPs. We do apologize for not having thought this through more carefully when revising the manuscript based on the reviewer's comments on the first version.
PLEASE SEE ALL THE CHANGES IN TRACK CHANGES IN THE ATTACHED DOCUMENT